# Factors affecting calving to conception interval (days open) in dairy cows located at Dessie and Kombolcha towns, Ethiopia

**Migbnesh Yekoye Temesgen**[1☯], **Alula Alemayehu Assen**[1☯]*, **Tarekegn Tintagu Gizaw**[1], **Bethelehem Alemu Minalu**[1], **Anteneh Yenehun Mersha**[2]

1 School of Veterinary Medicine, Wollo University, Dessie, Ethiopia, 2 Pinnacle Private Limited, Addis Ababa, Ethiopia

☯ These authors contributed equally to this work.
* alulaalemayehu@gmail.com

**Data Availability Statement:** All relevant data are within the paper and its Supporting Information files.

## Abstract

This study was aimed at determining the median days of calving to conception interval (days open) and identifying the major risk factors determining the days open in dairy cows. Both retrospective and longitudinal studies were conducted on dairy cows located in the Dessie and Kombolcha towns of South Wollo Zone, northeast Ethiopia from August 2019 to August 2020. The Kaplan-Meier model of survival analysis was used to determine the median days open and compare the survival distribution of each level of explanatory variables likely to influence the days open of dairy cows. Accordingly, the overall conception rate of dairy cows was 44.7%. The median days open in the study was 154 days. There is a 16% probability of surviving (i.e, the probability that the conception event has not yet occurred) at the end of 210 days postpartum period. The Cox proportional hazard model was used to quantify the effect of each of the explanatory variables on the days open in the first 210 days postpartum. Factors that had a significant effect (p<0.05) on the calving to conception interval were the season of insemination, breeding system, calving to insemination interval, and herd milk yield level. However, the peripartum, postpartum disorders, and the town of the farm are not significant (p>0.05). Accordingly, cows inseminated in the autumn season (HR = 4.45), cows less than or equal to 85 days calving to insemination interval (HR = 2.41), artificially inseminated cows (HR = 1.45), and high herd milk yield cow had a higher probability of becoming pregnant. In conclusion, the management practices and breeding decisions seem to be important determinants to improve the conception rate or decrease the days open in dairy cows.

## Introduction

Reproductive performance is the major concern of the modern dairy industry worldwide by determining the profitability of a dairy farm [1]. Poor reproductive performance is one of the most common reasons for culling in dairy herds [2]. Because, it affects the amount of milk produced per cow per day of herd life, affects the longevity of the cow in the herd, and indirectly

**Funding:** The author(s) received no specific funding for this work.

**Competing interests:** The authors have declared that no competing interests exist.

influences herd replacement costs, breeding costs, and expenses for veterinary treatment and drugs [3].

Generally, it is believed that a calving interval of about one year is an optimal indicator of fertility and profitability of dairy herds [4]. To obtain this calving interval, a postpartum cow has to restart ovarian activity, be detected in heat, be mated, and conceive within 85 days after calving, while producing large quantities of milk [5, 6]. The biological possibility of a new conception at this time after parturition is based on a coordinated working together of the hypothalamus, pituitary, ovaries, and uterus, resulting in an excellent uterine involution and an early resumption of ovarian function. Together with the efficacy of heat detection, the management decision on when to start breeding and the overall conception rate are the most important factors determining the length of the postpartum conception. In addition, diverse factors, e.g., body condition score (BCS) during transition or at service, heat stress, age or parity, milk yield, calving to first service interval, and peripartum disorders (dystocia, metritis, and retained placenta) have been interrogated for an effect on the calving-to-conception interval (days open) [5–7].

The calving-to-conception interval (days open), the period between parturition and the following conception of a dairy cow is the major parameter used to determine the reproductive performance and to make an economic decision in dairy herds [8]. Minimizing days open is economically beneficial by increasing milk yield relative to labor and feed costs [4, 9], increasing the number of calves [9] and lifetime productive days [10], as well as reducing breeding costs [9, 10] and culling rates [4, 9, 10]. For example, in the USA an average decrease of 2.40 kg of milk, 0.112 kg of fat, extended calving interval and decreased the number of calves available for replacement which account for an economic loss of about 0.25 to 0.71 USD cents for each additional day open [11].

Conception in a dairy cow is usually analyzed as a binary trait determined by whether or not a cow conceives after insemination following calving. Treating conception as a binary trait views pregnancy as having occurred during a defined period, hence ignoring the continuity of the conception process and the precise time of pregnancy or "failure" [12]. An analytical method such as survival analysis is now available that offers several advantages over standard regression techniques for quantifying days open as time intervals in dairy cattle [8, 13]. It accounts for continuity of the conception and does not restrict the analysis of the data to arbitrarily predefined time points. The main advantage of survival analysis is that it can retain the information from cows that are culled before conception or not pregnant by the time the data recording was completed. Thus, records from pregnant (uncensored) and non-pregnant (censored) cows can be treated jointly and included in the analysis, making proper use of all the available information. Within the field of fertility in dairy cattle, survival analysis has been applied to study the effects of diseases on days to conception [8, 14] the relationship between BCS and postpartum reproductive efficiency [15], and the effect of early lactation milk yield on days open [8].

There is a scarcity of research using survival analysis has been conducted for fertility traits particularly the calving to conception interval (days open) of dairy cows in dairy farms in the tropics, particularly in Ethiopia. Information about days open and relating factors is of paramount importance to dairy farmers as well as to extension agents, veterinarians, and researchers. Moreover, it can help to develop strategies of possible interventions for improving the reproductive performance of dairy cows in general and days open in particular. Therefore, the objective of this study was to determine the median days open and identify the major risk factors determining the days open in dairy cows of the study area.

## Materials and methods

### Study area

The study was conducted in Kombolcha and Dessie towns of South Wollo Zone, northeast Ethiopia from August 2019 to August 2020. Kombolcha and Dessie towns are situated in 11.08˚N, 39.72˚E and 11.13˚N, 39.14˚E, respectively. The altitudes are ranged from 1842–1915 and 2470-2553m above sea level for Kombolcha and Dessie towns, respectively. The mean annual rainfall of Kombolcha and Dessie towns ranged from 725.1 to 1361.6mm and 851.3 to 1612.6mm, respectively. The mean annual temperature varies from 18.7 to 20.9˚c and 14.8 to 19.3˚c at Kombolcha and Dessie, respectively [16]. The average seasonal temperature and humidity values during the study period were 19.7˚C, 64%; 21.7˚C, 63.3%;22.7˚C, 56.7% and 20.7˚C, 65% for winter, summer, spring and autumn, respectively. The towns are selected since dairy crossbreeding is widely promoted by the regional government through the distribution of pregnant crossbred heifers and the use of artificial insemination. In addition, the selected areas are potentially favorable for dairy and other livestock by the Amhara Regional State. Furthermore, the selected areas have high dairy cattle and human populations where many people have a habit to consume milk [17].

### Study animals

The study animals are crossbred dairy cows which were found in 61 dairy farms. A Crossbreeding scheme in Ethiopia and the study area is practiced to combine superior hardiness, heat tolerance, disease resistance, and environmental adaptability of indigenous cattle with superior high milk yield, faster growth rates and early maturity of temperate breeds, mainly Holstein Friesian cattle breed. In this study, a dairy herd size larger than 5 dairy animals were subjected to participate. In addition, dairy herds were selected based on their proximity to the road, management system, availability of dairy records, and the willingness of the farmer to participate. Accordingly, 61 dairy herds were included in this study where 21 and 40 herds were located in peri-urban and urban areas of Dessie and Kombolcha towns, respectively. A total of 385 cows in the dataset where 118 and 267 cows were from Dessie and Kombolcha towns, respectively. These animals were selected based on their parturition and health status during pre(post)partum period. Accordingly, cows that gave birth less than 45 days at the initial visit and whose disease history and date of the last calving known were recruited, retrospectively. The dairy production system of the follow-up animals was classified under periurban and urban milk production system where the herd is dominated with improved/crossbreed dairy cattle; the production system is market-oriented and milk production is mainly for sale [18].

### Sampling technique and study design

A purposive sampling technique was employed to select dairy farms and cows in the study area. Accordingly, the farms were considered based on herd size, accessibility, and availability of recording of the herd, etc. Both retrospective and prospective cohort studies were employed to collect data on the day's open of a cow.

### Study methodology

**Data source.** The data comprised 385 crossbreed cows in 61 purposively selected dairy herds in Dessie and Kombolcha towns, Ethiopia. These data were collected from August 2019 to August 2020. The dairy herds selected for this study were a convenience sample drawn

based upon ease of access to the data required for this study and the willingness of the herd owner to participate in the study. Purchased or entrusted cows were not included in this study.

**Data collection.** Each participating farm was visited by a team of research data collectors every 2 wk from August 2019 until August 2020. Sampled dairy farms were enrolled and followed for approximately 7 months/210 days. Dairy cows that gave birth less than 45 days at the initial visit and whose disease history and date of the last calving known were recruited retrospectively (concurrent cohort) and allowed to join into the prospective cohort. Other cows were recruited prospectively as they give birth within the selected farms during the study period. The recruited cows and those cows that gave birth after the initial visit were identified and recorded at the earliest farm visit. All study cows were regularly visited twice per month until the cows reached 7 months of postpartum. Cows are withdrawn from the follow-up when they completed their 7 months of postpartum. When cow loss happened during the follow-up period, the date and reason for the loss were recorded [8].

The response variable of interest is the calving-to-conception interval which is called days open (DO) in the first 210 days after calving [1]. In the study period, if a cow did conceive, she was considered to be uncensored. In contrast, if a cow did not conceive, she was considered to be censored. A censored observation meant that the event of interest (e.g., conception) had not occurred by the end of the study or by the time the subject was no longer observed (e.g., culling or dying) [8, 19]. This information showed that the event of interest had not occurred during the time that the subject was in the study. Cows that became pregnant during the study period were uncensored because conception was the outcome of interest. For these cows, days open (number of days between calving and subsequent conception) was the outcome variable. If the cow died or was sold and was not pregnant, the number of days postpartum at the last date of selling or death was the outcome variable. If the cow doesn't show the event of interest (e.g., conception) by the end of the study period, the number of days postpartum at the last date of the study period was the outcome variable. All cows that show the event was coded one, while those with incomplete records are coded as zero (i.e., pregnant cows = 1, and non-pregnant cows = 0).

Biological (herd milk yield, peri- and postpartum disorders, and calving to insemination interval) and environmental (herd, season of insemination) factors limiting conception [20] were recorded for each cow by the members of the research team and directly by farmers. Records of calving date, parity, peri(post)partum disorder, body condition score after calving and milk yield were recorded for cows recruited in the follow-up at each initial farm visit. Afterward, while farm visits were carried out every two weeks the date of insemination after calving, breeding system and date of pregnancy diagnosis and result were recorded.

**Variable definition.** The definition of variables included in the statistical model is presented below:

Survival time: For the survival analysis, the survival time (the interval between the starting point and the end of follow-up) is an outcome variable. In this study, the starting point was d0 after calving. The end of follow-up was defined as either the conception date, date of censoring (i.e. culling, death, loss and conception hadn't occurred by the end of the study) of the animal. Therefore, the values of survival time for days open (DO) were obtained from the interval between calving and the end of follow-up [15]. The endpoint of the DO was d210. Consequently, values of DO, higher than 210, were set equal to 210. A cow that did not conceive within 210 days but was still present was a censored observation.

Event status: All cows that show the event of interest (i.e conception) were recorded as pregnant cows while those with incomplete records and couldn't become pregnant were recorded as non-pregnant.

Seasons of insemination: defined according to rainfall profiles for the regions where the herds were located. The season of insemination category (i.e., SI) was defined within every

dairy cow by classifying the cow into four classes according to the season of insemination in the first 210 days after calving. Cows in class 1 were those inseminated in the autumn (September–November) season, cows in class 2, cows in class 3 and cows in class 4 were those inseminated in the winter (December- February), spring (March-May), and summer (June–August) seasons, respectively. This classification was intended to stratify the cows according to possible differences in seasonal and herd management effects for days open during the insemination season to look at possible nonlinear effects.

Herd milk yield (H_MY): the category was defined by classifying the herds into two classes according to the median 100-d milk yield. Accordingly, the median H_MY at 100d of lactation was 15 liters per day. Herds in class $H\_MY_1$ were those above the median, and herds in class $H\_MY_2$ were those located below the median. This classification was intended to stratify the herds according to differences in genetic level for milk yield or differences in management during lactation. For the calculation of milk yield at 100 d of lactation, cows were required to have at least 1 test day record in the periods 5 to 50, 50 to 100, and 100 to 150 d after calving [21].

Peri- and postpartum disorders: the definitions used in the present study were similar to those described previously [21–23]. Calving difficulty was ranked according to the degree of assistance required (1 = no assistance, 2 = minor assistance, 3 = some force required, 4 = significant force required, and 5 = cesarean section). Cows with a calving score >2 were considered to have dystocia. The retained placenta was defined as the retention of the fetal membranes for longer than 24 hours. Septicemic metritis was defined by the presence of fever (≥39.5˚C) and a watery, fetid uterine discharge during the first 10 days postpartum. Ketosis was diagnosed by the following clinical signs within 4 weeks postpartum: anorexia, depression, and the odor of acetone on the breath. Milk fever was diagnosed by the presence of weakness and recumbency after calving. Abomasal displacement was diagnosed by the detection of a 'ping' sound during abdominal auscultation within 4 weeks postpartum. Clinical endometritis was diagnosed based on the presence of a visible mucopurulent vaginal discharge and/or rectal palpation of the enlarged uterus at 4 weeks postpartum.

Calving to insemination interval: The time between calving and insemination of a cow. The variable was grouped into two classes considering 85 days as the optimal fertility indicator: short (< = 85 days) and long (> 85 days) as described by [5, 6].

## Data management and analysis

The collected data was stored in the Microsoft excel database system for data management. Statistical analyses were conducted using R version 3.6.1 [24]. The median days open for cows that did conceive was determined by the Kaplan-Meier model of the survival analysis method using the {survival} package of R [25]. In addition, the association between each explanatory variable thought to influence days open was tested using the log-rank test. Each explanatory variable was categorized into 2 or more levels. The Kaplan-Meier survival curves for each level of an explanatory variable were plotted using {GGally} package, which is an extension to the {ggplot2} package of R [26].

A Cox proportional hazard model [27] was fitted to the data to quantify the effect of explanatory variables on days open in the first 210 days postpartum. Accordingly, the full model was fitted to the data including all the explanatory variables such as the season of insemination, calving season, breeding system, calving to insemination interval, and herd milk yield level, and the interactions between these variables were included in the model. Backward stepwise regression technique was used to build a multivariable cox proportional hazard model, and elimination was performed based on the Wald statistic criterion when p>0.1 [28]. The final

model included the effects under analysis and all other significant main effects. The coefficient estimates of the survivor function and hazard ratios and 95% CI were obtained for all classes within factors included in the final model. The model is represented as follows:

$$\lambda(\mathbf{t}, \mathbf{X}) = \lambda_0(\mathbf{t})e^{\beta_1 x1 + \beta_2 x2 + \cdots \beta_p xp} \tag{1}$$

Where, $\lambda(\mathbf{t,x})$ = hazard of an event for a cow at time t with covariate x,

$\lambda o(\mathbf{t})$ = baseline hazard function describing the hazard of an event for a hypothetical situation when all covariate values are set to zero, and

$e^{\beta_1 x1 + \beta_2 x2 + \cdots \beta_p xp}$ = term specific to individuals with covariate x1, x2. . .xp.

A p-value $< 0.05$ was considered statistically significant.

The set of covariates x are:

$SI_i$ = fixed effect of season of insemination i in which the cow was last inseminated (i = 1 to 4),

$BS_k$ = fixed effect of the breeding system used for cow k (k = 1 to 2),

CIl = fixed effect of calving to insemination interval category l (l = 1 to 2),

H_MYm = fixed effect of herd milk yield category m (m = 1 to 2),

$TOWN_n$ = fixed effect of the origin (town) of the cow n (n = 1 to 2),

PRE. DIS.$_o$ = fixed effect of prepartum disorder of the cow o (o = 1 to 2),

POST. DIS.$_p$ = fixed effect of postpartum disorder of the cow p (p = 1 to 2) and

## Ethics statement

This study was carried out in cooperation with dairy farmers in the study areas and formalized by an official letter of cooperation obtained from the Research and Community Service Vice-President Office of Wollo University, Ethiopia. The purpose of the study was explained to all dairy farms included in the study and informed consent was sought from farm owners willing to be included in the study. The study was carried out under the existing dairy farm setting and the selection of dairy cows was for analysis cohorts. All interactions with the cows in this study were part of routine farm management where the researcher's involvement in this study was only observing and recording data in routine farm processes. Approval from an Institutional Animal Care and Use Committee (IACUC), or equivalent animal ethics committee, was not obtained as the researchers didn't manipulate cows or the experimental design led to differential treatment from routine practices on farms.

## Results

### Kaplan-Maier modeling of days open

The Kaplan-Maier estimate of the median days open in the study cow was 154 days with 95% CI (150.06, 157.94). The conception rate of dairy cows in the study was 44.7%. In addition, from the total 385 dairy cows followed in the study, 172 (44.7%) and 213 (55.3%) observations were uncensored and censored observations by 210 days postpartum, respectively. The minimum and maximum censoring time of the study were 2 and 210 days; whereas 79 and 191 days were for the uncensored time of the study animals, respectively (Table 1).

A plot of Kaplan-Meier survival functions of the probability of postpartum conception is shown in Fig 1. The survival curves showed that there is a 16% probability of surviving (i.e, the probability that conception has not yet occurred) at the end of 210 days postpartum period. The survival probability starts to fall after d110 after calving.

**Table 1. Descriptive statistics of calving to conception interval (days open) under analysis.**

| Observations | Days open |
|---|---|
| Total records, no. | 385 |
| Censored records, no | 213 |
| Minimum censoring time, day | 2 |
| Maximum censoring time, day | 210 |
| Median censoring time, day | 159 |
| Uncensored records, no. | 172 |
| Minimum failure time, day | 79 |
| Maximum failure time, day | 191 |
| Median failure time, day | 154 |

The results of the log-rank (Mantel-Cox) test of equality of the survival distributions for the different levels of factors on days open of dairy cows were shown in Table 2. Accordingly, factors that had a significant association (p< 0.05) with days open are the season of insemination, calving season, breeding system, and calving to insemination interval. However, herd milk yield, town, and pre(post)partum disorder are not significant (p> 0.05).

## Cox proportional-hazards modeling of days open

The coefficient estimates of the survivor function and hazard ratios (HR) and 95% CI of the HR were obtained for all levels of factors included in the final model fitted to the multivariable Cox proportional hazard model is presented in Table 3. Accordingly, the insemination season, calving season, breeding system, calving-to-insemination interval, and herd milk yield level have shown significant effects (p<0.05) on the days open of dairy cows.

The median days open for cows inseminated in summer, autumn, winter, and spring was found to be 169, 142, 154, and 151 days, respectively (Table 2). A significant difference (p<0.05) was found among cows inseminated in different seasons (Table 2). The estimated

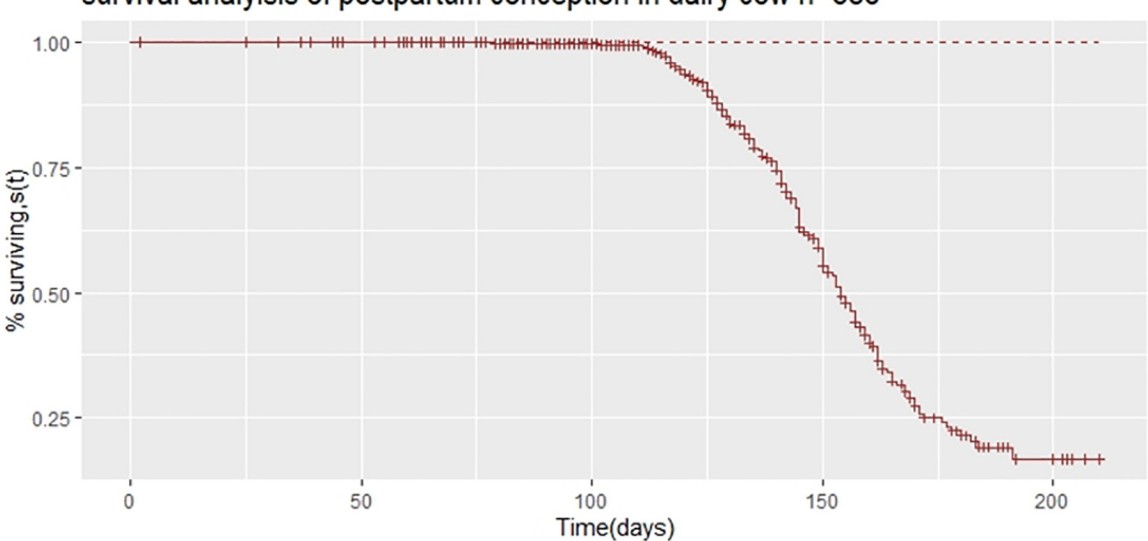

**Fig 1. Kaplan-Meier estimates of survivor function of days open in dairy cows.**

**Table 2. Log-rank (Mantel-Cox) test of equality of the survival distributions for the different levels of factors on days open of dairy cows.**

| Factor | Level | Median (SE) | 95% CI | $X^2$ | p-value |
|---|---|---|---|---|---|
| **Insemination season** | Summer | 169 (4.35) | (160.48, 177.52) | 61.19 | 0.000 |
| | Autumn | 142 (4.12) | (133.93, 150.07) | | |
| | Winter | 154 (4.02) | (146.11, 161.89) | | |
| | Spring | 151 (4.05) | (143.06, 158.94) | | |
| **Breeding system** | AI | 152 (2.27) | (147.54, 156.46) | 5.26 | 0.022 |
| | NS | 159 (3.45) | (152.23, 165.77) | | |
| **Calving to Insemination interval** | < = 85 days | 144 (1.29) | (141.48, 146.52) | 41.87 | 0.000 |
| | > 85 days | 165 (3.20) | (158.73, 171.27) | | |
| **Herd milk yield (H_MY)** | H_MY1 | 155 (2.04) | (151.00, 158.99) | 1.14 | 0.286 |
| | H_MY2 | 152 (2.69) | (146.72, 157.28) | | |
| **Town** | Dessie | 152 (1.94) | (148.20, 155.80) | 1.27 | 0.260 |
| | Kombolcha | 156 (2.91) | (150.06, 157.94) | | |
| **Prepartum disorder** | No | 154 (2.06) | (149.96, 158.03) | 0.19 | 0.661 |
| | Yes | 154 (12.25) | (129.99, 178.00) | | |
| **Postpartum disorder** | No | 153 (1.80) | (149.47, 156.53) | 0.49 | 0.484 |
| | Yes | 158 (2.50) | (153.10, 162.89) | | |

SE is the standard error of the median estimate.

χ2 is the chi-square statistics.

CI is the 95% confidence interval of the estimate.

NS is Natural service.

AI is Artificial Insemination.

hazard ratios show a linear effect of the season of insemination on days open. Cows inseminated in autumn (HR = 4.45), winter (HR = 2.12), and spring (HR = 1.93) had a higher probability of becoming pregnant than in summer (Table 3). The survivor function for days open was plotted for cows categorized by the season of insemination (Fig 2). The plot revealed that

**Table 3. Multivariate Cox proportional hazards model of days open of dairy cows in Dessie and Kombolcha.**

| Factors | Level | β (SE) | HR (95%CI) | p-value |
|---|---|---|---|---|
| **Season of insemination** | Summer | | 1 | 0.000 |
| | Autumn | 1.49 (0.22) | 4.45 (2.88, 6.87) | |
| | Winter | 0.75 (0.21) | 2.12 (1.39, 3.23) | |
| | Spring | 0.66 (0.26) | 1.93(1.15,3.25) | |
| **Breeding system** | NS | | 1 | 0.042 |
| | AI | 0.37 (0.18) | 1.45 (1.01, 2.01) | |
| **Calving-to-insemination interval** | >85 days | | 1 | 0.000 |
| | < = 85 days | 0.88 (0.17) | 2.41 (1.73, 3.36) | |
| **Heard milk yield (H_MY)** | H_MY2 | | 1 | 0.016 |
| | H_MY1 | -0.46 (0.19) | 0.63 (0.44,0.92) | |

β is regression parameter of the survivor function.

SE is the standard error of the median.

HR is the Hazard ratios within variable are given relative to the hazard for the reference class, which is set to 1.0.

CI is the 95% confidence interval.

NS is Natural service.

AI is Artificial Insemination.

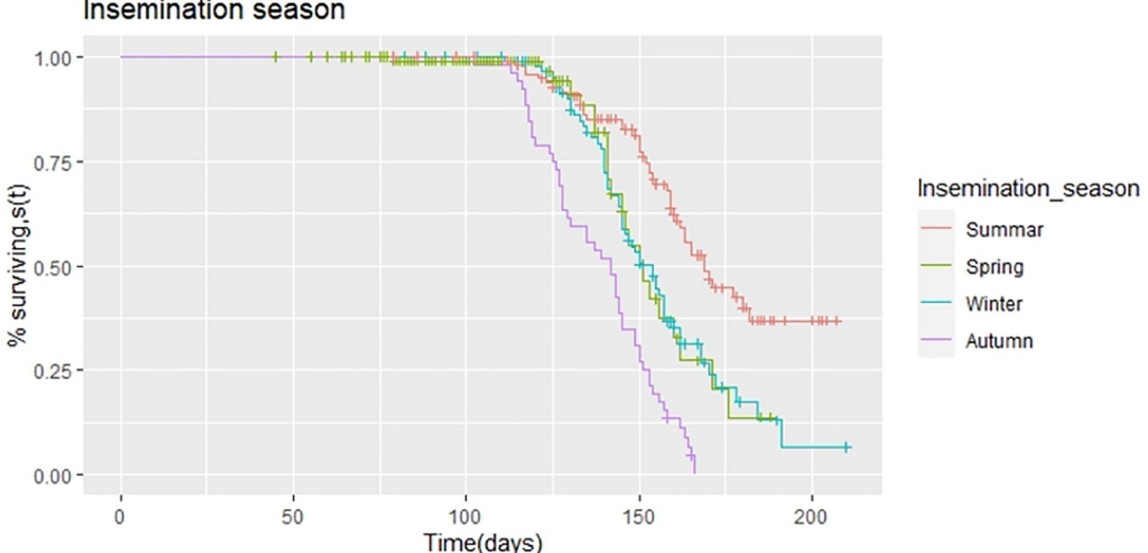

**Fig 2. Kaplan-Meier estimates of survivor function for the days open of dairy cows with the season of insemination category.**

the survival curve for cows inseminated in autumn is consistently smaller than survival curves for cows inseminated in other seasons, respectively. For almost any day open, the survival probability (i.e., the cows do not have a record of conception) is higher for cows inseminated in the summer season. In other words, the chance of a failure (i.e., the cow reaches conception) is lower for cow inseminated in the summer season. The survival probability of cows inseminated during the winter and spring seasons are almost equivalent. This result is in agreement with the results obtained with the hazard ratios.

The median days open in cows with less than or equal to 85 days and greater than 85 days calving to insemination interval were 144 and 165 days, respectively (Table 2). This showed a 21 days extension of days open in cows that have greater than 85 days calving to insemination interval than in cows that account less than or equal to 85 days calving to insemination interval. There was a significant difference (p<0.05) among calving to insemination interval strata on days open (Table 2). Accordingly, cows that have less than or equal to 85 days calving to insemination interval had a higher probability (HR = 2.41) to conceive than cows that have greater than 85 days calving to insemination interval (Table 3). The survival function curve of days open among the calving to insemination interval factor is presented in Fig 3. This plot reveals that the survival curve for cows with greater than 85 days calving to insemination interval strata is consistently higher than survival curves for cows under the strata less than or equal to 85 days calving to insemination interval. The main differences appear only after 100 d from the calving date.

The median days open for natural service (NS) exposure was 159 days, compared to 152 days open for Artificial insemination (AI) breeding system exposed cows (Table 2). There was a 7 days extension in days open for cows with natural service relative to the artificial insemination breeding system. A significant difference (p<0.05) was found among the breeding system used on days open. Compared to cows exposed to natural services, cows exposed to artificial insemination were 1.45 times more likely to become pregnant or to have decreased days open (Table 3). The association between the breeding system used on days open of dairy cows is demonstrated using the survival function curve (Fig 4). The plot shows that the survival curve for cows using natural service (i.e bull) is higher than survival curves for cows using an artificial insemination breeding system. The main differences appear only after around 130 d from the calving date.

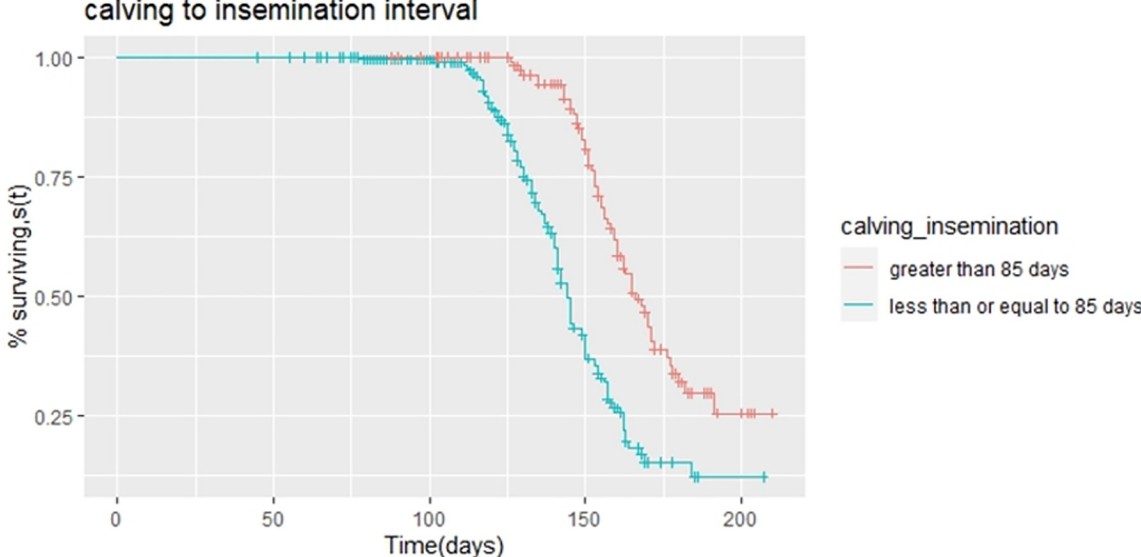

**Fig 3. Kaplan-Meier estimates of survivor function for the days open with calving to insemination category.**

There was no significant difference (p>0.05) between herd milk yield and days open (Table 2). However, a significant effect (p<0.05) of herd milk yield on days open adjusted for other covariates in the multivariable Cox proportional hazard model. Estimates of hazard ratios for herd milk yield indicate that cows in the low milk yield category were 0.63 times as likely to become pregnant as were cows from herds in the high category (Table 3). Thus, cows from herds with a higher milk yield category have more chance of getting pregnant.

## Discussion

The overall median days open obtained in this study was 154 days. This can produce a calving interval of fewer than 400 days. However, it is not comparable to a calving interval of 12–13

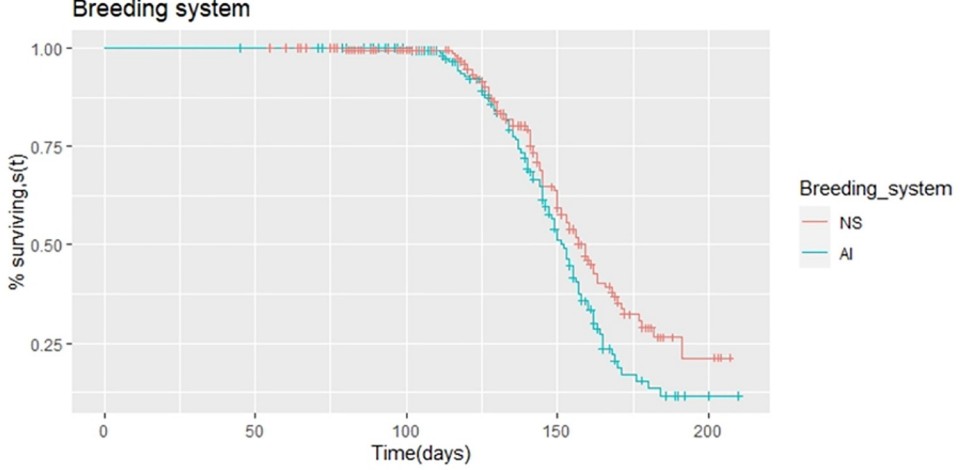

**Fig 4. Kaplan-Meier estimates of survivor function for days open with the breeding system used category.**

months, which is suggested as optimal for dairy herds [29]. The median days open found in this study is higher than 109.8 days in Costa Rica [21] and 131 days in Thailand [30], Vietnam, China, and Myanmar (110–118 days) and is comparable with countries such as Bangladesh, Pakistan, Sri Lanka, and Indonesia (147–255 days) [30]. The variation in days open in dairy cattle is mainly determined by factors operating at either the individual cow or lactation level. In addition, the differences could have been by the failure of the dairy farmer to detect heat signs after calving. Thus, the calving to insemination interval was prolonged and eventually influenced the days open.

The median days open of dairy cows inseminated in the different seasons were found to be 142, 154, 151, and 169 days for cows inseminated in autumn, winter, spring, and summer, respectively. The days open were significantly differing ($p < 0.05$) among the insemination season of cows where the shorter days open was found in autumn inseminated cows than cows inseminated in other seasons, which is in agreement with the finding of [31]. This might be attributed to the positive influence of the autumn season on the reproductive performance of cows via optimal environments and the availability of fresh green pastures through the early lactation period [31] and an improved health condition could explain this finding. Cows inseminated in the summer season are less likely to be pregnant than other seasons of the year. According to weather condition of the study area, the summer season mainly June and July are the warmest months of the year with a monthly average temperature ranging from 73 to 84°F. As a result, the high temperature is an important determinant of reproductive performance by decreasing the intensity and duration of estrus that leads to an increased incidence of estrus detection failures. This in turn reduces both the total number of inseminations and the proportion of inseminations that result in pregnancy [32]. Heat stress alters concentrations of circulating hormones by increasing circulating concentrations of corticosteroids [33] and by reducing progesterone concentration [34, 35]. The viability of pre-fixation embryos is reduced [36], and the uterine environment is altered by decreased blood flow [37] and increased uterine temperature [38]. Although cows are not typically classified as seasonal breeders, changes in photoperiodic stimulation [39] associated with specific times of the year provide a potential explanation for such an effect of season on the reproductive measures [40].

The calving to insemination interval was proved to be an important index for evaluating postpartum conception in dairy cows. In this study, the calving to insemination interval factor has significantly ($p < 0.05$) affected days open of dairy cows. Accordingly, the dairy cow with smaller calving to insemination interval (less than or equal to 85 days) is more likely (HR = 2.41) to become pregnant than cows with large calving to insemination interval (> 85 days). The results from other studies suggest a curvilinear response with conception rates lower amongst cows where calving to service interval is very long [21, 41, 42], which agrees with our finding. This may be due to selection biases with other factors amongst cows whose first service was delayed causing lower conception rates. Although delaying the first service may increase conception rates, this strategy may improve primary measures of reproductive performance. Contrary to our results, the conception rate was lower in cows with short calving to insemination interval in a Spanish study [43]. These discrepancies might be due to the different categories of the interval from calving to insemination used in each study.

Our findings suggest that herd milk yield levels resulted in considerable effect ($p < 0.05$) on days open (Table 3). The estimated hazard ratios indicate that cows in the low category were less likely (HR = 0.63) to be pregnant than were cows from herds in the high category (Table 3). This finding contradicts Eicker et al. [44] who has shown that the effect of milk yield on conception rate is minimal. This might be due to the level of milk yield in that study being considered a continuous variable. In addition, Herman et al. [8] revealed a higher chance of conception for heifers from herds with lower milk yield. In theory, cows with a high milk yield

are likely to have more days open because of the negative effect of milk yield on energy balance and reproductive performance. Our result does not fully support this effect because cows in the highest milk yield category showed a higher chance to become pregnant, and the difference among the herd milk yield level was not clear. The lower conception rate in low-producing cows might be caused by proper management decisions on the farm. Milk yield is an important factor in breeding decisions on most dairies [44]. Farmers likely do not show the same interest in breeding low-yielding cows.

Artificially inseminated cows had a higher probability of becoming pregnant than cows exposed to natural service. These results of higher hazard rates for pregnancy in AI exposed cows are consistent with the report of Overton and Sischo [45]. This result is not unforeseen as many dairy farms with bullpens may move cows that experience calving-related trauma and reproductive tract injuries directly into bullpens, or will move them when cows are believed to have fertility problems. Dairy farmers reported that cows are moved from AI into bullpens when the cow is a repeat breeder, extended days in milk (DIM), and poor milk production. This suggests that the dairy producers believe that bull breeding improves the likelihood that these non-pregnant and potentially lower producing cows will become pregnant, and that the natural service (NS) breeding system lowers their input costs to these less productive cows [45].

The days open of dairy cows in Dessie and Kombolcha town of the study area was not significantly different (p>0.05) which is probably because the areas have similar nutritional and health management and productivity of cows [1].

The effect of pre(post)partum disease in this study was inconclusive; although a significant effect (p>0.05) of pre(post)partum disease was not observed which is likely due to the small number of cows with reported pre(post)partum disease.

## Conclusions

This study aims to determine the median days open and identify the major risk factors affecting the days open in dairy cows. The season of insemination, calving season, breeding system, the calving to insemination interval, and herd milk yield level appears to have a significant effect (p<0.05) on days open. As a result, dairy cows calved and inseminated in the autumn and winter season of the year were more likely to be pregnant or decrease the days open, respectively. In addition, cows with shorter calving to insemination interval and artificial insemination exposed tend to have a greater probability to conceive than cows with longer calving to insemination interval and natural service exposed cows. Herd milk yield seemed to affect the days open. Herds with higher milk yield had a higher chance of being pregnant; however, management practices and breeding decisions seem to be more important for the situation analyzed here because the herds with the lowest yield had the lowest chance of getting pregnant. Based on the finding of this study, dairy farmers should have proper management practices and breeding decisions that are important determinants to improve the conception rate of the dairy cow.

## Supporting information

**S1 Dataset.**
(XLS)

**S1 File. R script used for analysis.**
(R)

## Acknowledgments

The authors gratefully acknowledge dairy farm owners at Dessie and Kombolcha towns for allowing us to conduct this study and follow-up their animals during the study period.

## Author Contributions

**Conceptualization:** Alula Alemayehu Assen.

**Data curation:** Migbnesh Yekoye Temesgen, Tarekegn Tintagu Gizaw, Bethelehem Alemu Minalu, Anteneh Yenehun Mersha.

**Formal analysis:** Alula Alemayehu Assen.

**Investigation:** Migbnesh Yekoye Temesgen, Tarekegn Tintagu Gizaw, Bethelehem Alemu Minalu.

**Methodology:** Alula Alemayehu Assen.

**Project administration:** Alula Alemayehu Assen.

**Software:** Alula Alemayehu Assen.

**Supervision:** Alula Alemayehu Assen, Tarekegn Tintagu Gizaw.

**Validation:** Alula Alemayehu Assen, Tarekegn Tintagu Gizaw, Bethelehem Alemu Minalu, Anteneh Yenehun Mersha.

**Visualization:** Alula Alemayehu Assen, Tarekegn Tintagu Gizaw, Anteneh Yenehun Mersha.

**Writing – original draft:** Migbnesh Yekoye Temesgen, Alula Alemayehu Assen.

**Writing – review & editing:** Alula Alemayehu Assen, Tarekegn Tintagu Gizaw, Bethelehem Alemu Minalu, Anteneh Yenehun Mersha.

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
