## [Decision Letter · Decision Letter 0]

29 Nov 2021

PONE-D-21-31364Survival analysis of calving to conception interval in dairy cows located at Dessie and Kombolcha towns, EthiopiaPLOS ONE

Dear Dr. Assen,

Thank you for submitting your manuscript to PLOS ONE. After careful consideration, we feel that it has merit but does not fully meet PLOS ONE’s publication criteria as it currently stands. Therefore, we invite you to submit a revised version of the manuscript that addresses the points raised during the review process.

We look forward to receiving your revised manuscript.

Kind regards,

Dawit Tesfaye

Academic Editor

PLOS ONE

Journal Requirements:

2. In your Methods section, please provide additional location information, including geographic coordinates of your field collection site if available.

Reviewers' comments:

Reviewer's Responses to Questions

**Comments to the Author**

1. Is the manuscript technically sound, and do the data support the conclusions?

Reviewer #1: Yes

Reviewer #2: Yes

2. Has the statistical analysis been performed appropriately and rigorously? 

Reviewer #1: Yes

Reviewer #2: Yes

3. Have the authors made all data underlying the findings in their manuscript fully available?

Reviewer #1: No

Reviewer #2: Yes

4. Is the manuscript presented in an intelligible fashion and written in standard English?

Reviewer #1: Yes

Reviewer #2: Yes

5. Review Comments to the Author

Reviewer #1: Dear Authors

Many thanks for your interesting work.

In general you have applied appropriate statistical methods and models to your data in a clear way and understandable way. Although these statistical approaches are not new, in my opinion your results are important for countries with similar climate and production conditions like Ethiopia. However, I need more information about the conditions (production system, climate …) of your study. Regarding these aspects, I have ranked your work into the category “Major revisions”.

Details, which needs further clarifications:

L54: You have postulated that an acceptable CI is about 85 days. Why did you use a threshold of 80 days in your analysis?

L65: Please provide the literature source.

L98: Although it might be difficult to answer but important: What are the breeds used in the crossbred scheme? Where are the herds located? How many herds were in one town? ...

L102ff: How many herds are within the dataset? Please give more information about the production system and herd sizes.

L113: I have problem to understand what you mean by “censored”.

a) Did you integrate only cows, which have the chance to become pregnant? In other words, cows with a calving date close to the final (max) date of your dataset were discarded?

b) If a cow was culled within the 210 day period because of low milk yield is also coded as 0? Do you have reliable information about this?

It might be that you have answered these questions in the lines L121ff. However, I have problems to understand. Please clarify.

L147: Your initial model should include interaction terms, but I could no find any information about this in the rest of your paper.

L157: I am wondering that you integrate two seasonal factors (insemination, calving season) into your model which might be confounded. I mean, if you look at the distribution (4 x 4 table) of your data, most information should be on the diagonal. If this is true, you should skip one of the effects from your model.

L179: “This classification … genetic level …” I do not understand this sentence. Needs some clarification.

L181: Please provide some information about the threshold in units of milk yield within 100d.

L256: Please check the numbering of you figures.

L296: “The variation in days … or lactation level”. I do not understand this sentence. Needs some clarification.

L300: It would be useful if you could provide some information about temperature and humidity of the seasons. This would help to understand your results.

Reviewer #2: 1. Keywords in the abstract part should be put in alphabetical order and contain key words

2. The introductory part should review variable under study and also should have gap that lead the researcher to conduct this work.

3. The research topic (title) and the objective (s) should match!!

4. In the methodology part the following questions should be answered!

a. Why only the two towns from South Wollo?

b. The number of dairy farms from each town & how these farms were selected?

c. The number of animals from each town & how these animals were selected?

d. How sample size was determined?

e. Sources of data and method of data collection were poorly indicated!!!

f. Variables to be measured should be clearly put

g. Production system should get due attention!!!!!!!!!!

5. Tables in the result part should be clearly put and justified.

6. The major points from this findings in each table should be well discussed. Few points from the table were put in the form of conclusion!!!

7. The figures are not witnessing the discussed points in the body part.

8. The discussion part seems well written

9. The manuscript lacks recommendation

10, References should be listed in alphabetical order!

11. Generally it is advisable to follow the article writing manual

6. PLOS authors have the option to publish the peer review history of their article (what does this mean?). If published, this will include your full peer review and any attached files.

Reviewer #1: No

Reviewer #2: No

---

## [Author Response · Author response to Decision Letter 0]

25 Jan 2022

- File naming was edited to comply with the style requirements. We hopefully have no divergences from the PLOS ONE’s style requirements now.

2. In your Methods section, please provide additional location information, including geographic coordinates of your field collection site if available.

- Thank you for the comment. The suggestions made by the editor were included in the revised manuscript from L102 to L108.

- The dataset and R script are provided as part of the manuscript supporting information. In addition, the dataset and R script used for analysis are available at the figshare repository via the following link https://doi.org/10.6084/m9.figshare.19062554.v1

- Thank you for the suggestions given we have reviewed the references used in the body of text and the lists of references thoroughly and ensured that it is complete and correct. We haven’t cited a retracted paper. However, in the revised manuscript we have included new reference lists to substantiate newly included pieces of information and removed references for literature that are omitted in the revised manuscript. 

Reviewers comments

Reviewer 1:

Reviewer #1: Dear Authors

Many thanks for your interesting work.

In general you have applied appropriate statistical methods and models to your data in a clear way and understandable way. Although these statistical approaches are not new, in my opinion your results are important for countries with similar climate and production conditions like Ethiopia. However, I need more information about the conditions (production system, climate …) of your study. Regarding these aspects, I have ranked your work into the category “Major revisions”.

- We thank you very much for the positive compliment about the work and its importance for countries like Ethiopia. We agreed that we have missed information on the climate and production system of the area from the manuscript. Hence, we have included this information in the revised manuscript from L108 to L110 and L130 to L134, respectively.

Details, which needs further clarifications:

L54: You have postulated that an acceptable CI is about 85 days. Why did you use a threshold of 80 days in your analysis?

- Thank you for your suggestions and comments forwarded. We are sorry for the discrepancies created and we admitted that the threshold value used in our analysis and introduction should be inline. Hence, we have done the analysis again considering 85 days as a threshold cutoff value for calving to insemination interval. Thus, we stratified the calving to insemination interval (CI) variable which is continuous data type into two classes (less than or equal to 85 days vs greater than 85 days) to look at possible nonlinear effects. 

L65: Please provide the literature source.

- Thank you for the comment and we amend the statement to some extent and included the literature source in L74.

L98: Although it might be difficult to answer but important: What are the breeds used in the crossbred scheme? Where are the herds located? How many herds were in one town? ...

- Thank you very much for the concern and suggestions. All the information missed from the previous manuscript was included from L117 to L126.

L102ff: How many herds are within the dataset? Please give more information about the production system and herd sizes.

- We are grateful for the comments forwarded. The number of dairy herds included in the dataset, herd sizes and production system of the herds were included in the revised manuscript from L124 to L125.

L113: I have problem to understand what you mean by “censored”.

- We have elaborated what censored observation meaning in the revised manuscript from L64 to L167.

a) Did you integrate only cows, which have the chance to become pregnant? In other words, cows with a calving date close to the final (max) date of your dataset were discarded?

- Cows with known last calving date and which are not yet pregnant will be included in the follow up (retrospectively or prospectively). Then, during the follow up period, the cow may show the event of interest (i.e. conception) considered as an event and other cows may not show the event of interest due to death, culling or sold and end of the study period are considered as censored observation. Therefore, the result of the follow up study was either conception (event) or not (censored). Thus, every cow has a chance to be included in the follow-up although the study period was close to the final date of your dataset. As a result, we haven’t discard cows with a calving date close to the final date of follow-up period.

b) If a cow was culled within the 210 day period because of low milk yield is also coded as 0? Do you have reliable information about this? It might be that you have answered these questions in the lines L121ff. However, I have problems to understand. Please clarify.

- Yes, if a cow under study is culled from the herd due to poor characteristics such as low milk yield is considered as censored observation (coded as 0). We can have reliable information on each and every cow under follow-up since we made a regular herd visit on their pregnancy status every 2 weeks and collect the required information.

L147: Your initial model should include interaction terms, but I could no find any information about this in the rest of your paper.

- Thank you for the comments. Since we didn’t include and compute the interaction term in the full model we have removed these ststistics in the revised manuscript.

L157: I am wondering that you integrate two seasonal factors (insemination, calving season) into your model which might be confounded. I mean, if you look at the distribution (4 x 4 table) of your data, most information should be on the diagonal. If this is true, you should skip one of the effects from your model.

- Thank you for the concern and suggestions. We have removed one of the possible confounding variables (calving season), which have a relationship with insemination season and event of interest (i.e. conception).

L179: “This classification … genetic level …” I do not understand this sentence. Needs some clarification.

- We are sorry for the lack of clarity we produced and made improvements to elaborate further in the revised version of the manuscript from L207 to L209 and hopefully this wording clarify better than the previous one.

L181: Please provide some information about the threshold in units of milk yield within 100d.

- Thank you for the suggestions and comments. We have made the improvement on the comments from L211 to L215. 

L256: Please check the numbering of you figures.

- Thank you for the comments and we have checked and corrected. 

L296: “The variation in days … or lactation level”. I do not understand this sentence. Needs some clarification.

- Edited in the revised manuscript

L300: It would be useful if you could provide some information about temperature and humidity of the seasons. This would help to understand your results.

- Thank you for the suggestions and we have included the required information from L108 to L110

Reviewer #2: 

1. Keywords in the abstract part should be put in alphabetical order and contain key words

- Thank you for the suggestion. We put the keywords in alphabetical order in the revised manuscript and included keywords in L43.

2. The introductory part should review variable under study and also should have gap that lead the researcher to conduct this work.

- We have included additional literature on the variable in introduction section from L54 to L64

3. The research topic (title) and the objective (s) should match!!

- We thank you for the suggestions and amend the title of the manuscript as “Factors affecting calving to conception interval (Days open) in dairy cows located at Dessie and Kombolcha Towns, Ethiopia” in the revised manuscript in L1 to L3. Therefore, we amended the title of the manuscript from the electronic submission form. 

4. In the methodology part the following questions should be answered!

a. Why only the two towns from South Wollo?

- The reason why only the two towns selected from south wollo zone were included in the revised manuscript from L110 to L115. 

b. The number of dairy farms from each town & how these farms were selected?

- The information lacked on the number of dairy farms from each town and how they are selected is included in the revised manuscript from L121 to L126.

c. The number of animals from each town & how these animals were selected?

- Thank you very much for the suggestions and we have included the required information in the revised manuscript from L126 to L130.

d. How sample size was determined?

- Thank you for the question and concern. We couldn’t calculate the sample size of this study because there is no formula for setting the sample size for the purposive sampling technique employed in this study. There is no need for a statistical representative sample. Any number of sample (sample size) can be selected, which can serve the purpose of the researcher. Instead, judgments must be made, based on the expected heterogeneity of areas, population groups, locations, households and individuals. 

e. Sources of data and method of data collection were poorly indicated!!!

- We thank you for the suggestions and we are sorry for the lack of clarity created. The data source and data collection employed were included in the revised manuscript from L142 to L184.

f. Variables to be measured should be clearly put

- the variables measured and recorded in the study were presented under the variable definition subtopic in the revised manuscript from L187 to L234.

g. Production system should get due attention!!!!!!!!!!

- Thank you for the suggestion and we have incorporated the dairy production system in the revised manuscript from L131 to L134.

5. Tables in the result part should be clearly put and justified.

We thank you for the suggestions and improved the tables in the revised manuscript.

6. The major points from this findings in each table should be well discussed. Few points from the table were put in the form of conclusion!!!

We have described the tables better in the revised manuscript. 

7. The figures are not witnessing the discussed points in the body part.

We agreed on the comment and included the description of each figure in the revised manuscript.

8. The discussion part seems well written

We are thankful for the positive compliment.

9. The manuscript lacks recommendation

Thank you for the suggestions. We have included the recommendation in the revised manuscript from L457 to L459.

10, References should be listed in alphabetical order!

Thank you for the suggestions and we have arranged the references as per the PLOS ONE journal guideline. It says the reference list should be put as per the order of the reference in the body of the text. As a result, I listed the reference accordingly.

11. Generally it is advisable to follow the article writing manual

We have admitted the concern and suggestions from the reviewer and the corresponding author and co-authors believed that the revised manuscript is prepared in accordance with the PLOS ONE journal.

---

## [Editor Report · Decision Letter 1]

2 Feb 2022

Factors affecting calving to conception interval (days open) in dairy cows located at Dessie and Kombolcha towns, Ethiopia

PONE-D-21-31364R1

Dear Dr. Assen,

We’re pleased to inform you that your manuscript has been judged scientifically suitable for publication and will be formally accepted for publication once it meets all outstanding technical requirements.

Kind regards,

Dawit Tesfaye

Academic Editor

PLOS ONE
---

## [Editor Report · Acceptance letter]

8 Feb 2022

PONE-D-21-31364R1 

Factors affecting calving to conception interval (days open) in dairy cows located at Dessie and Kombolcha towns, Ethiopia 

Dear Dr. Assen:

I'm pleased to inform you that your manuscript has been deemed suitable for publication in PLOS ONE. Congratulations! Your manuscript is now with our production department. 

Kind regards, 

on behalf of

Dr. Dawit Tesfaye 

Academic Editor

PLOS ONE